# CCAAT/Enhancer-Binding Protein β (C/EBPβ) Regulates Calcium Deposition in Smooth Muscle Cells

**DOI:** 10.3390/ijms252413667

**Published:** 2024-12-20

**Authors:** Nakwon Choe, Sera Shin, Young-Kook Kim, Hyun Kook, Duk-Hwa Kwon

**Affiliations:** 1Department of Pharmacology, Chonnam National University Medical School, Hwasun 58128, Republic of Korea; 2Basic Research Laboratory for Vascular Remodeling, Chonnam National University Medical School, Hwasun 58128, Republic of Korea; 3BK21 Plus Center for Creative Biomedical Scientists, Chonnam National University, Gwangju 61186, Republic of Korea; 4Department of Biochemistry, Chonnam National University Medical School, Hwasun 58128, Republic of Korea

**Keywords:** vascular calcification, vascular smooth muscle cells, C/EBPβ, Runx2, Runx2 P2 promotor

## Abstract

Calcium deposition in vascular smooth muscle cells (VSMCs), a form of ectopic ossification in blood vessels, can result in rigidity of the vasculature and an increase in cardiac events. Here, we report that CCAAT/enhancer-binding protein beta (C/EBPβ) potentiates calcium deposition in VSMCs and mouse aorta induced by inorganic phosphate (Pi) or vitamin D_3_. Based on cDNA microarray and RNA sequencing data of Pi-treated rat VSMCs, C/EBPβ was found to be upregulated and thus selected for further evaluation. Quantitative RT-PCR and Western blot analysis confirmed that C/EBPβ was upregulated in Pi-treated A10 cells, a rat VSMC line, as well as vitamin D_3_-treated mouse aorta. The overexpression of C/EBPβ in A10 cells increased bone runt-related transcription factor 2 (*Runx2*), alkaline phosphatase (*ALP*), and osteopontin (*OPN*) mRNA in the presence of Pi, as well as potentiating the Pi-induced increase in calcium contents. The Runx2 expression was increased by C/EBPβ through Runx2 P2 promotor. Our results suggest that a Pi-induced increase in C/EBPβ is a critical step in vascular calcification.

## 1. Introduction

Vascular calcification is a process characterized by the abnormal deposition of calcium phosphate crystals in blood vessels [1] and is widespread in individuals with chronic diseases such as diabetes, chronic kidney disease, and atherosclerosis and aging populations [2]. Vascular calcification contributes to increased arterial stiffness, reduced vascular compliance, and impaired blood flow, thus increasing the risk of cardiovascular events, organ damage, and mortality [3].

Vascular smooth muscle cells (VSMCs) play a crucial role in maintaining vascular structure and function, regulating vascular tone, blood pressure, and blood flow [4]. However, VSMCs can undergo phenotypic changes and differentiate into osteoblast-like cells, leading to vascular calcification [5]. This pathological process is induced by various factors such as elevated levels of inorganic phosphate (Pi), inflammatory cytokines, and oxidative stress [6].

CCAAT/enhancer-binding protein beta (C/EBPβ) is a C/EBP family transcription factor involved in various biological processes including cell survival apoptosis [7], tumorigenic transformation [8], inflammation, and metabolism [9]. It plays critical roles in conditions such as pneumonia, osteoarthritis, and hepatitis, as well as in processes like B cell, adipocyte, myeloid, and ovarian luteal cell differentiation [10]. C/EBPβ is expressed in a multitude of tissues and cell types, including the heart [11], brain [12], liver, adipose tissue [13], bone marrow cells, and immune cells [14], and its expression is modulated by various stimuli, including cytokines, hormones, and cellular stresses [15].

Recent studies have revealed the role of C/EBPβ in bone and cartilage development and calcification. C/EBPβ has been shown to promote the expression of genes involved in matrix mineralization and hypertrophic differentiation in chondrocytes [16]. Additionally, in osteoblasts, C/EBPβ has been found to regulate the expression of osteogenic transcription factors such as runt-related transcription factor (Runx2) and activating transcription factor 4 (ATF4) [17]. It also regulates osteogenic markers, including osteocalcin and alkaline phosphatase (ALP), thereby contributing to bone formation [18]. In mice, the loss of C/EBPβ results in reduced bone mass due to decreased osteoblast differentiation and function. Furthermore, C/EBPβ deficiency retards growth during embryogenesis and postnatal stages, owing to inhibited chondrocyte differentiation [17]. However, the role and mechanism of C/EBPβ in VSMC calcification largely remain to be investigated.

Here, we investigated the potential role of C/EBPβ in the progression of vascular calcification. We observed the C/EBPβ expression in VSMCs stimulated with Pi and in the aortas of rats treated with vitamin D_3_, two well-established models of vascular calcification [19,20,21], as well as the effects of over-expression and downregulation on VSMC calcification. Our findings indicate that the expression of C/EBPβ is increased in response to Pi and vitamin D_3_, and the C/EBPβ expression is correlated to VSMC calcification.

## 2. Results

### 2.1. C/EBPβ Is Upregulated in Inorganic Phosphate with VSMCs

To investigate the potential involvement of C/EBPβ in vascular calcification, we first examined its expression in two established models of this pathological process. The treatment of rat VSMCs with Pi led to a time-dependent increase in C/EBPβ mRNA levels, as assessed by both RNA sequencing and microarray (GSE 130486) (Figure 1a). This observation was further confirmed in A10 cells, a rat VCMC cell line, by quantitative RT-PCR, which showed a significant upregulation of C/EBPβ mRNA at days 3 and 6 of Pi treatment (Figure 1b). The increase in C/EBPβ protein expression was also checked by Western blot analysis (Figure 1c,d). Collectively, these results demonstrate that C/EBPβ expression is significantly upregulated in Pi-induced VC, suggesting its involvement in the progress of VC.

### 2.2. C/EBPβ Is Increased in Vitamin D_3_-Induced VC Mice

To determine whether C/EBPβ is regulated in VC mouse models, we induced VC in mice through the subcutaneous injection of vitamin D_3_ (VD_3_) for 3 consecutive days, followed by a maintenance period. Nine days after the initial VD_3_ injection, aortic tissues were isolated and analyzed. The administration of VD_3_ to mice resulted in a significant increase in C/EBPβ mRNA levels in the aorta (Figure 2a). This upregulation was accompanied by a marked increase in C/EBPβ protein expression, as demonstrated by Western blot analysis (Figure 2b,c). Together, these results suggest that elevated C/EBPβ expression in VSMC is linked to vascular calcification and may contribute to its progression as a pathological process.

### 2.3. C/EBPβ Overexpression Elevates Calcium Deposition

We next investigated the role of the C/EBPβ in calcium deposition in VSMCs. The overexpression of C/EBPβ (Figure 3a) significantly enhanced Pi-induced calcification, as evidenced by increased Alizarin Red S staining (Figure 3b), elevated calcium content (Figure 3c), and upregulation of the calcification-related genes, including *Runx2*, *ALP*, and *OPN*, as well as the matrix-related gene, collagen type 1 alpha 1 (*Col1a1*) (Figure 3d).

Conversely, the knockdown of C/EBPβ using siRNA (Figure 4a) attenuated Pi-induced calcification, as demonstrated by decreased Alizarin Red S staining (Figure 4b), lower calcium content (Figure 4c), and reduced expression of the calcification-related genes, such as *Runx2*, *ALP*, and *OPN*, along with the matrix-related gene, *Col1a1* (Figure 4d). These findings indicate that C/EBPβ is both necessary and sufficient for the full extent of Pi-induced calcification in VSMCs.

### 2.4. C/EBPβ Regulates Runx2 P2 Promoter Activity in VSMCs

To investigate the molecular mechanisms underlying the pro-calcific effects of C/EBPβ, we examined its impact on the expression of Runx2, a key transcription factor driving osteogenic differentiation and calcification [22]. Runx2 expression is under control of two different promoters: the distal P1 and the proximal P2 promoter. The type II Runx2 driven by the P1 promoter is predominantly expressed in bone progenitor cells and is upregulated during bone formation. In contrast, the type I Runx2 driven by the P2 promoter is expressed in both osteogenic and non-osteogenic cells [23]. We explored whether C/EBPβ regulates VC by activating the Runx2 promoter. The overexpression of C/EBPβ in VSMCs significantly enhanced the activity of the Runx2 P2 promoter in the presence of Pi, while it downregulated the activity of the Runx2 P1 promoter (Figure 5a,b). These findings suggest that the Runx2-I isoform, controlled by the P2 promoter, may play a more critical role in VSMC calcification and that its expression is positively regulated by C/EBPβ in response to calcification-inducing stimuli.

## 3. Discussion

The present study explores a vascular calcification mechanism focusing on C/EBPβ. Through in vitro and in vivo models, we investigated the function of C/EBPβ in driving the progression of vascular calcification. C/EBPβ expression is upregulated in response to calcification-inducing stimuli, leading to the enhancement of the Runx2 promoter, which in turn upregulates ALP and OPN mRNA levels, promoting the process of VC (Figure 5c). These results demonstrate that C/EBPβ plays a critical role in promoting calcification in vascular smooth muscle cells (VSMCs) and the mouse aorta.

It is evident that C/EBPβ, a transcription factor from the CCAAT/Enhancer Binding Proteins (C/EBPs) family, exerts diverse regulatory influences across various cellular contexts. Particularly, C/EBPβ is instrumental in the early differentiation stages of coronary smooth muscle cells (SMCs) from proepicardial origins, where its presence in the nucleus is crucial. This factor also facilitates the differentiation of 10T1/2 fibroblasts into SMCs [24] and VSMC proliferation by EGFR-dependent activation [25], as established in foundational studies. Intriguingly, Yang et al. demonstrated that C/EBPβ, in conjunction with POU class 2 homeobox 2 (POU2F2), directly binds to the promoter region of the vasoactive factor endothelin 1 (EDN1), modulating its expression in prehypertensive SHR vascular smooth muscle cells [26]. This modulation is critical for the structural remodeling of these cells.

Furthermore, the interaction of C/EBPβ with ATF4 in the context of osteoblast differentiation in VSMCs highlights a pathway leading to vascular calcification through increased phosphate uptake [27]. Conversely, the overexpression of C/EBPβ in porcine aortic valve interstitial cells (pAVICs) during calcific nodule formation represses canonical Wnt signaling, effectively preventing calcific nodules [28]. This suggests a protective role against calcification under these specific conditions.

However, contrasting observations from Masuda et al. [27], alongside our findings, suggest a pro-calcifying influence of C/EBPβ in vascular contexts. The discrepancy in actions between vascular calcification and valvular calcification mediated by C/EBPβ underscores the complexity of its biological functions and necessitates further investigation. Understanding the differential impacts of C/EBPβ in these calcification processes could illuminate new therapeutic targets for calcific diseases, reflecting the nuanced interplay of transcriptional regulation by C/EBPβ across different cellular environments.

We also monitored the effect of C/EBPβ on the Runx2 promoter, a key transcription factor driving osteogenic differentiation and calcification [22], and found that C/EBPβ differentially regulates the activity of Runx2 P1 and P2 promoters in VSMCs. Runx2 has two distinct promoters: the distal P1 and the proximal P2 promoter. The Runx2 P1 promoter drives the type II transcript, encompassing the exon 1–8, and primarily regulates bone formation [29], in contrast to the Runx2 P2 promoter, which drives the type I transcript, encompassing exon 2–8, and governs both osteogenic and non-osteogenic processes [30]. In this study, the overexpression of C/EBPβ downregulated the activity of the Runx2 P1 promoter, while significantly enhancing the activity of the Runx2 P2 promoter in the presence of Pi. This finding is particularly intriguing because previous studies have demonstrated that C/EBPβ binds to the P1 promoter of the Runx2 gene and upregulates its expression in other types of cells including osteoblasts [31].

The differential regulation of Runx2 promoters by C/EBPβ in VSMCs suggests that the transcriptional control of Runx2 expression may be cell type dependent. In osteoblasts, C/EBPβ has been shown to cooperate with other transcription factors, such as specificity protein 1 (SP1) and erythroblast transformation-specific proto-oncogene 1 (ETS), to co-stimulate the Runx2 P1 promoter [18]. However, in VSMCs, our data indicate that C/EBPβ activates the Runx2 P2 promoter while downregulating the P1 promoter. To our knowledge, this is the first study to report the differential regulation of the Runx2 promoter by C/EBPβ in VSMCs, highlighting the potential importance of the Runx2 P2 isoform in the context of vascular calcification. While previous studies have shown that C/EBPα regulates VSMC calcification [32] and co-localizes with Runx2 in chondrocytes [33], the specific regulation of Runx2 promoters by C/EBPβ in the context of VSMC calcification has not been previously investigated.

The distinct roles of Runx2 P1 and P2 isoforms in osteoblast differentiation and bone development have been well documented [29,33,34]. However, their specific functions in the context of vascular calcification remain largely unexplored. Our findings suggest that the Runx2 P2 isoform may be more relevant to VSMC calcification and that its expression is positively regulated by C/EBPβ in response to calcification-inducing stimuli such as Pi.

Given the clinical significance of vascular calcification in the context of aging and chronic diseases [2], our results highlight the potential of C/EBPβ as a novel therapeutic target for preventing or treating this pathological process. Further research is required to investigate potential crosstalk with other signaling pathways involved in calcification and assess the translational potential of targeting C/EBPβ for the prevention or treatment of vascular calcification in clinical settings.

In conclusion, our study provides compelling evidence that C/EBPβ plays a critical role in promoting vascular calcification in response to calcification-inducing stimuli, at least in part by specifically activating the Runx2 P2 promoter in VSMCs. These novel findings might contribute to our understanding of the transcriptional regulation of vascular calcification and highlight the potential of C/EBPβ as a therapeutic target for the pathological process.

## 4. Materials and Methods

### 4.1. Small Interfering RNA (siRNA), and Antibodies

C/EBPβ siRNA, and control siRNA (scramble) were obtained from Bioneer Corp (Daejeon, Republic of Korea). The following antibodies were used at a 1:1000 dilution: anti-C/EBPβ (sc-7962, Santa Cruz Biotechnology, Dallas, TX, USA), anti-Runx2 (23981, Abcam, Cambridge, UK), and anti-β-Actin (sc-47778, Santa Cruz Biotechnology).

### 4.2. Cell Cultures

In this study, 6~7-week-old male Sprague Dawley rats were anesthetized with 2,2,2-Tribromoethanol (240 mg/kg; intraperitoneal injection) (T48402, Sigma-Aldrich, St. Louis, MO, USA) before isolating the thoracic aorta. After being rinsed with phosphate-buffered saline (PBS), the aorta was incubated in Ham’s F12 medium (12-615F, Lonza, Alpharetta, GA, USA) containing 0.2% collagenase I (LS004196, Worthington, Lakewood, NJ, USA) in a 37 °C incubator for 30 min. The intima was scraped from the luminal surface of the longitudinally opened aorta and then cut in Ham’s F12 media supplemented with 300 U/mL penicillin and 300 U/mL streptomycin. The minced sample was further digested in a 0.2% collagenase I solution in a 37 °C incubator for 30 min. The isolated VSMCs from rats were cultured in DMEM (LM001-05, Welgene, Gyeongsan, Republic of Korea) containing 10% fetal bovine serum (FBS, S001-07, Welgene) and antibiotics (15240062, ThermoFisher Scientific, Waltham, MA, USA). The cells at passage 2 to 6 were used for experiments.

A10 cells (CRL-1476, from American Type Culture Collection, Manassas, VA, USA), derived from the thoracic aorta of embryonic rats and resembling myoblasts, were employed as a model system of rat VSMCs. The A10 cells were cultured in DMEM supplemented with 10% FBS. All cells were maintained at 37 °C in a humidified incubator with 5% CO_2_.

### 4.3. Induction of Vascular Calcification

VC was induced in cultured cells by adding 2 mM or 4 mM Pi to the culture medium. The medium was replaced every 2 days, and the cells were treated for a total duration of either 4 or 6 days.

For inducing VC in mice models, 8- to 9-week-old male C57BL/6 mice were administered with VD_3_ solution subcutaneously at a dose of 150 μL/25 g (equivalent to 5 × 10^5^ IU/kg/day) for three consecutive days, followed by a 6-day maintenance period to promote VC [20,35]. VD_3_ solution was prepared by dissolving 14.56 mg of cholecalciferol (C9759, Sigma-Aldrich) in 70 μL of 100% ethanol followed by mixing with cremophor (Alkamuls EL-620, Sigma-Aldrich) for 15 min at room temperature. The mixture was then combined with 6.2 mL of distilled water containing 250 mg of dextrose for an additional 15 min at room temperature. Mice were euthanized using CO_2_ inhalation to evaluate aortic calcification.

### 4.4. mRNA Microarray and RNA Seq Analysis

An mRNA microarray (Agilent Microarray, Agilent-028282, Santa Cruz Biotechnology) was performed after the pooling of 3 samples. RNA was isolated as described below. Two pooled samples were used for the microarray independently to reduce the experimental error, and the averaged values were further evaluated. The results were previously submitted to the Gene Expression Omnibus (GEO) database and are accessible under accession code GSE74755 [36].

For RNA-Seq analysis, total RNA was extracted using the TRIzol reagent (ThermoFisher Scientific), and the Rbo-Zero Gold rRNA Remval kit (Illumina, Agilent, San Diego, CA, USA) was used to remove rRNA. The RNA sequencing library was prepared with the TruSeq Stranded Total RNA Kit (Illumina). RNA-sequencing analysis was performed using the HiSeq 2500 (Illumina, Agilent) library in the paired-end mode.

### 4.5. Quantification of Calcium Deposition

Cells were washed twice with PBS and then decalcified by incubating them in 0.6 N HCl overnight at 4 °C. The content of calcium in the HCl supernatant was measured using the QuantiChrom^TM^ Calcium Assay Kit (DICA-500, BioAssay Systems, Hayward, CA, USA). Samples were combined with working reagent, and the absorbance was measured at 570 nm using the ELx808 Absorbance Reader (BTELX808, BioTek Instruments, Winooski, VT, USA). The remaining cells were lysed in a solution of 0.1 N NaOH and 0.1% SDS, and the protein concentration was determined using the BCA Protein Assay kit (23225 ThermoFisher Scientific). Calcium levels were normalized to the protein content.

### 4.6. Quantitative Real-Time Polymerase Chain Reaction (qRT-PCR)

Total RNA was isolated using the NucleoSpin^®^ RNA/Protein (740933.250, Macherey-Nagel, Düren, Germany), according to the manufacturer’s instructions. Reverse transcription of mRNA into cDNA was performed using the SuperScript™ First-Strand Synthesis System for RT-PCR (11904018, ThermoFisher Scientific). The resulting cDNA was analyzed by qRT-PCR using the QuantiTect SYBR Green PCR Kit (204141, Qiagen, Hilden, Germany) on a Rotor-gene Q real-time PCR cycler (9001550, Qiagen, Hilden, Germany). Either Gapdh or 18S RNA served as the internal control. A pre-designed qPCR primer for Runx2 (Rn01512298_m1), ALP (Rn00677879_m1), and 18sRNA (Rn03928990_g1) were purchased from ThermoFisher Scientific. The sequences of primers used for PCR are as follows: C/EBPβ, forward, 5′-CCAAGAAGACGGTGGACAA-3′, and reverse, 5′-TTGCGCATCTTGGCCTT-3′; OPN, forward, 5′-CCGTGAGGCCGCAGTTCTCC-3′, and reverse, 5′-CAGAGGGCACGCTCAGACGC-3′; Col1a1, forward, 5′-AGCATGTCTGGTTTGGAGAG-3′, and reverse, 5′-GTGATAGGTGATGTTCTGGGAG-3′; GAPDH, forward, 5′-TGCACCACCAACTGCTTA G-3′, and reverse, 5′-GATGCAGGG ATGATGTTC-3′.

### 4.7. Western Blot Analysis

Cellular proteins were extracted using RIPA buffer (EBR001-500, Enzynomics, Daejeon, Republic of Korea) supplemented with 1 mM dithiothreitol (DTT), 1 mM sodium phosphate Na_3_PO_4_, 1 mM phenylmethylsulfonyl fluoride, and a protease inhibitor cocktail (11697498001, Hoffmann-La Roche, Basel, Switzerland). Proteins were separated using sodium dodecyl sulfate–polyacrylamide gel electrophoresis (SDS-PAGE) and transferred to a polyvinylidene difluoride (PVDF) membrane (Millipore, Bedford, MA, USA). The membranes were blocked with 5% skim milk (232100, BD Difco, Franklin Lakes, NJ, USA) in Tris-buffered saline containing Tween-20 (1xTBXT, 20605, ThermoFisher Scientific) (1TBST) and incubated with primary antibodies overnight at 4 °C on a rocker. After being washed three times in 1xTBST, membranes were treated with horseradish peroxidase (HRP)-conjugated secondary antibodies (7076 and 7074, Cell Signaling, Danvers, MA, USA) for 1 hour at room temperature. Chemiluminescent signals were detected using Western blotting luminol Reagent (sc-2048, Santa Cruz Biotechnology) and visualized with a FUJIFILM Luminescent Image Analyzer LAS-3000 (Fujifilm Life Science, Richmond, VA, USA). Band density was quantified using Scion Image software 4.0.3.2 (Scion Corporation, Frederick, MD, USA).

### 4.8. Cloning

The coding sequence of rat C/EBPβ was inserted into the pcDNA6/myc-His vector (V22120, ThermoFisher Scientific) to enable the overexpression of C/EBPβ in A10 cells. Additionally, a DNA fragment containing the Runx2 promoter sequence was subcloned into pGL3-Basic vector (E1741, Promega, Madison, WI, USA) for subsequent luciferase reporter assays.

### 4.9. Luciferase Assay

A10 cells were co-transfected with the Runx2 luciferase reporter vector and C/EBPβ expression construct using Lipofectamin 2000 reagents (ThermoFisher Scientific), following the manufacturer’s protocol. Luciferase activity was assessed using the Luciferase Assay System (E1500, Promega). To account for transfection efficiency, luciferase signals were normalized to the beta-galactosidase activity.

### 4.10. Statistical Analysis

All data are shown as mean ± S.E.M. Statistical analyses were performed using one-way ANOVA or Student’s t-tests, followed by Tukey’s post hoc test for multiple comparisons, where appropriate. Analyses were conducted using PASW Statistics 27 software (SPSS, an IBM Company, Chicago, IL, USA).

## Figures and Tables

**Figure 1 ijms-25-13667-f001:**
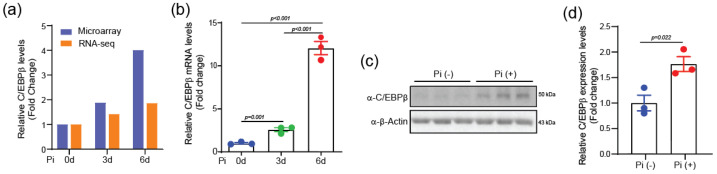
C/EBPβ is upregulated in VSMCs by treatment with inorganic phosphate. (**a**) Microarray and RNA-seq analyses revealed increased C/EBPβ expression in VSMCs exposed to Pi, with values averaged from two datasets. (**b**,**c**) C/EBPβ mRNA and protein levels were significantly elevated following 6 days of Pi treatment in A10 cells. *n* = 3. (**d**) Densitometry of chemiluminescence signal from the Western blot. Pi: inorganic phosphate. All data are presented as mean ± SEM. Statistical comparisons were performed using two-tailed *t*-test and ANOVA.

**Figure 2 ijms-25-13667-f002:**
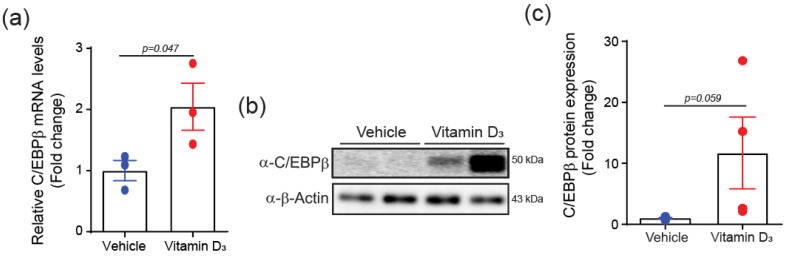
C/EBPβ is upregulated in mouse aorta by the administration of vitamin D_3_. (**a**,**b**) qPCR and Western blot analysis showing the increase in C/EBPβ transcript and protein content in response to vitamin D_3_ administration in mouse aorta. *n* = 3. (**c**) Densitometry of chemiluminescence signal from the Western blot. Data are presented as mean ± SEM. Statistical comparisons were performed using two-tailed *t*-test.

**Figure 3 ijms-25-13667-f003:**
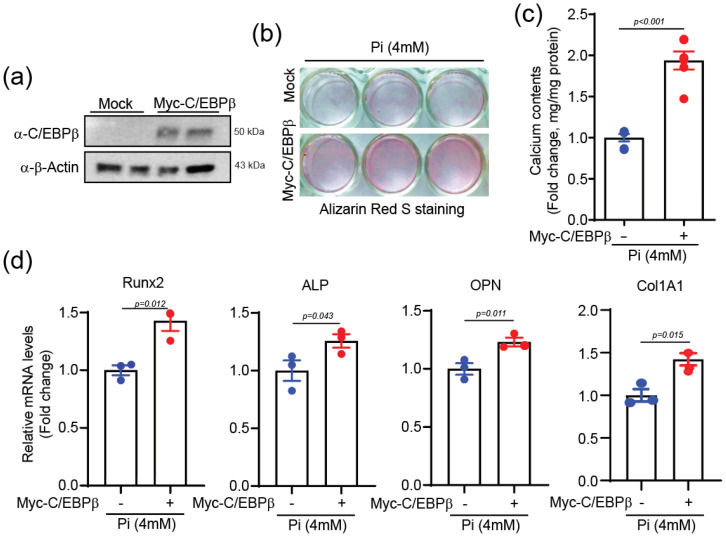
C/EBPβ potentiated VSMC calcification. (**a**) Overexpression of C/EBPβ in A10 was confirmed by Western blot. *n* = 2. (**b**) Alizarin Red S staining indicates increased calcification in A10 cells over expressing C/EBPβ cells. The day after the transfection, A10 cells were treated with 4 mM Pi for 4 days. (**c**) Increased calcification in A10 cells overexpressing C/EBPβ was quantified. *n* = 4. (**d**) qPCR results showing increase in calcification-related genes (*Runx2*, *ALP*, *OPN*) and matrix-related gene (*Col1a1*) in A10 cells overexpressing C/EBPβ. *n* = 3. Data are presented as mean ± SEM. Statistical comparisons were performed using two-tailed *t*-test.

**Figure 4 ijms-25-13667-f004:**
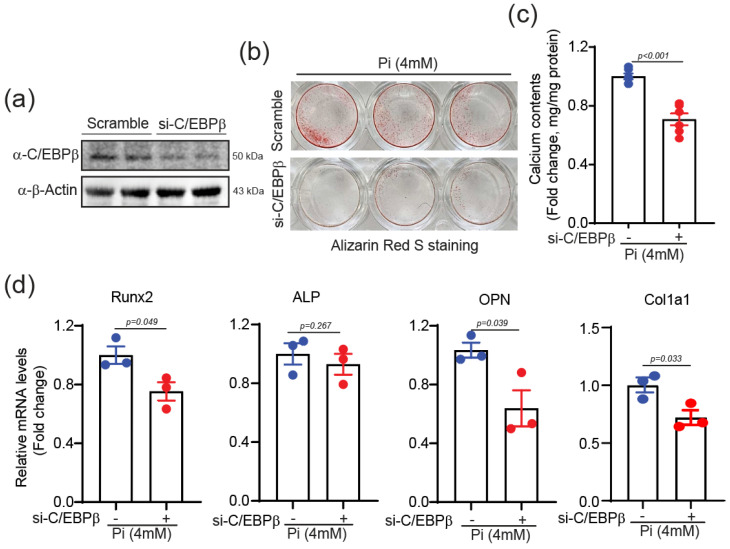
Knocking down C/EBPβ attenuated calcium deposition in VSMC. (**a**) Knocking down of C/EBPβ in A10 cells was confirmed by Western blot. *n* = 2. (**b,c**) Alizarin Red S staining and quantification of calcium deposition in Pi-treated A10 cells indicate attenuated calcification. A10 cells were transfected with C/EBPβ siRNA, and subsequently treated with 4 mM Pi for 4 days. *n* = 6. (**d**) qPCR results indicating knocking down of C/EBPβ in A10 cells resulted in downregulation of calcification-related genes (*Runx2*, *ALP*, *OPN*) and matrix-related gene (*Col1a1*). *n* = 3. Data are presented as mean ± SEM. Statistical comparisons were performed using two-tailed *t*-test.

**Figure 5 ijms-25-13667-f005:**
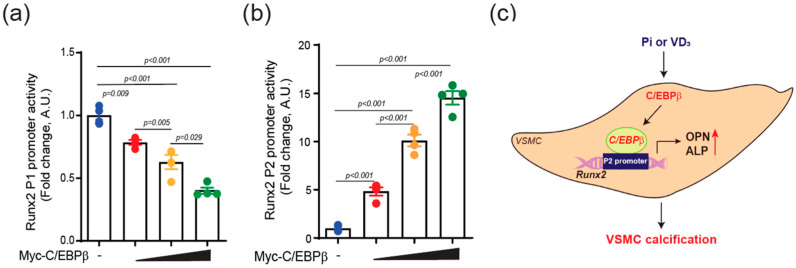
C/EBPβ modulates the activity of the Runx2 promoter. (**a**,**b**) Luciferase assay results showing the overexpression of C/EBPβ (100, 200, 400 ng) in A10 cells significantly enhanced the activity of the Runx2 P2 promoter, while it downregulated the activity of the Runx2 P1 promoter. *n* = 4. Data are presented as mean ± SEM. Statistical comparisons were performed using ANOVA. (**c**) Our results propose a role of C/EBPβ in VSMC calcification through the Runx2 promoter. Under calcification conditions, upregulated C/EBPβ activates the P2 promoter of Runx2, thereby inducing VC by increasing the mRNA levels of calcified genes such as ALP and OPN.

## Data Availability

The data that support the findings of this study are openly available in Gene Expression Omnibus (GEO) database with accession code GSE7477 at doi: 10.1111/jcmm.15670.

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
