# Peer review of "CCAAT/Enhancer-Binding Protein β (C/EBPβ) Regulates Calcium Deposition in Smooth Muscle Cells"

_ijms, 2024, doi:10.3390/ijms252413667_

Round 1
Reviewer 1 Report
Comments and Suggestions for Authors
Summary
This manuscript reports that CCAAT/enhancer-binding protein beta (C/EBPβ) promotes calcification of vascular smooth muscle cells (VSMCs), which is one of the causes of arteriosclerosis and subsequent cardiovascular disease. Although the results consistently demonstrated that C/EBPβ induces osteogenic gene expression and subsequent calcification of VSMCs, the main text is too short to fully describe the background, experimental objectives, methods, and detailed explanation of the results. The molecular mechanism of how C/EBPβ promotes the osteogenic program in VSMCs is still unclear. Animal experiments are also insufficient to support the hypothesis that C/EBPβ is involved in vascular calcification in vivo. This study only shows the correlation between the expression level of C/EBPβ and the vascular calcification index.
Comments
1. Lines 42-53: Basic and recent knowledges of C/EBPβ must be introduced in accurate. It is no information in “various biological processes”, “multiple tissues”, “various stimuli”. And the molecular mechanism of C/EBPβ-induced osteogenesis should be described.
2. Descriptions of the results are quite simple to explain the data.
3. Figure 2: It is not proved whether administration of vitamin D3 really caused vascular calcification in mice. I think it is impossible to induce vascular calcification by only three days of vitamin D3 treatment. The results just show that vitamin D3 increases the expression of C/EBPβ in the mouse aorta.
4. Figure 3: The days of osteogenic differentiation should be described.
5. Figures 3 and 4: Other matrix genes (collagen, osteocalcin, etc.) need to be quantified and analyzed to support the hypothesis.
6. Figure 5: Explain about P1 and P2 promoters. Figure 5C does not illustrate the difference between P1 and P2 promoters, how C/EBPβ works, why C/EBPβ induces RUNX2 expression, and how Pi induces C/EBPβ.
7. The sequences of qPCR primers should be provided.
Author Response
General comments
- Lack of detailed description and insufficient mechanistic details on how C/EBPβ promotes osteogenesis in VSMCs
This manuscript reports that CCAAT/enhancer-binding protein beta (C/EBPβ) promotes calcification of vascular smooth muscle cells (VSMCs), which is one of the causes of arteriosclerosis and subsequent cardiovascular disease. Although the results consistently demonstrated that C/EBPβ induces osteogenic gene expression and subsequent calcification of VSMCs, the main text is too short to fully describe the background, experimental objectives, methods, and detailed explanation of the results. The molecular mechanism of how C/EBPβ promotes the osteogenic program in VSMCs is still unclear. Animal experiments are also insufficient to support the hypothesis that C/EBPβ is involved in vascular calcification in vivo. This study only shows the correlation between the expression level of C/EBPβ and the vascular calcification index.
We sincerely thank the reviewer for their thoughtful feedback. In response, we have expanded the manuscript to include a more detailed description of the background, objectives, methods, and results. Your suggestions have greatly improved the clarity of our work.
Regarding the molecular mechanism of C/EBPβ in promoting osteogenesis in VSMCs, we acknowledge the importance of elucidating the intermediary pathways. While our study focused on RUNX2, a master regulator of VSMC osteogenic transformation, we agree that understanding the signaling pathways involved is critical. Ongoing research by our team is investigating RUNX2 promoter regulation and cofactors such as ATF4, interacting with C/EBPβ, which will be addressed in future studies.
Although the calcification observed with Vitamin D3 administration may appear stronger than clinical findings, this model is widely recognized for its reliability and reproducibility. Other methods, such as high-fat or calcium diets, resulted in only localized calcification, insufficient for biochemical analysis. Thus, we selected the Vitamin D3-induced model to ensure robust and repeatable results, as detailed in the following responses.
This study represents the first report of C/EBPβ’s effect on calcium deposition in VSMCs, providing a foundation for future mechanistic investigations. Thank you again for your invaluable comments.
Specific comments
- Provide accurate and updated information on C/EBPβ.
Lines 42-53: Basic and recent knowledges of C/EBPβ must be introduced in accurate. It is no information in “various biological processes”, “multiple tissues”, “various stimuli”. And the molecular mechanism of C/EBPβ-induced osteogenesis should be described.
As per your suggestion, we have provided a more detailed description of C/EBPβ and added a new paragraph to address its role in various biological processes, tissues, and stimuli.
2.1. various biological processes”, “multiple tissues”, “various stimuli
We revised the paragraph as following:
CCAAT/enhancer-binding protein beta (C/EBPβ) is a C/EBP family transcription fac-tor involved in various biological processes including cell survival, apoptosis [7], tumorigenic transformation [8], inflammation and metabolism [9]. It plays critical roles in conditions such as pneumonia, osteoarthritis, and hepatitis, as well as in processes like, B cell, adipocyte, myeloid and ovarian luteal cell differentiation [10]. C/EBPβ is expressed in multitude of tissues, and cell types including the heart [11], brain [12], liver, adipose tissue [13], bone marrow cells and immune cells [14]
2.2. the molecular mechanism of C/EBPβ-induced osteogenesis should be described.
We revised the paragraph as following:
Recent studies have revealed the role of C/EBPβ in bone and cartilage development and calcification. C/EBPβ has been shown to promote the expression of genes involved in matrix mineralization and hypertrophic differentiation in chondrocytes [16]. Additionally, in osteoblasts, C/EBPβ has been found to regulate the expression of osteogenic transcription factor such as runt-related transcription factor (Runx2) and activating transcription factor 4 (ATF4) [17]. It also regulates osteogenic markers, including osteocalcin and alkaline phosphatase (ALP), thereby contributing to bone formation [18]. In mice, the loss of C/EBPb results in reduced bone mass due to decreased osteoblast differentiation and function. Furthermore, C/EBPb deficiency retards growth during embryogenesis and postnatal stages, owing to inhibited chondrocyte differentiation [17]. However, the role and mechanism of C/EBPβ in VSMC calcification largely remain to be investigated.
2.3. Descriptions of the results are quite simple to explain the data.
Followings were revised as highlighted
Results 2-1
Collectively, these results demonstrate that C/EBPβ expression is significantly upregulated in Pi-induced VC, suggesting its involvement in the progress of VC.
Results 2-2
To determine whether C/EBPβ is regulated in VC mouse models, we induced VC in mice through subcutaneous injection of vitamin D3 (VD3) for 3 consecutive days, followed by maintenance period. On 9 days after the initial VD3 injection, aortic tissues were isolated and analyzed.
Results 2-3
Runx2 expression is under control of two different promoters: the distal P1 and the prox-imal P2 promoter. The type II Runx2 driven by P1 promoter is predominantly expressed in bone progenitor cells and is upregulated during bone formation. In contrast, the type I Runx2 driven by P2 promoter is expressed in both osteogenic and non-osteogenic cells [23]. We explored whether the C/EBPβ regulates VC by activating the Runx2 promoter.
- Figure 2: It is not proved whether administration of vitamin D3 really caused vascular calcification in mice. I think it is impossible to induce vascular calcification by only three days of vitamin D3 treatment. The results just show that vitamin D3 increases the expression of C/EBPβ in the mouse aorta.
Thank you for your insightful comment. To address your concern, we have added more details in the Methods section regarding the administration of Vitamin D3 (VD3) and the induction of vascular calcification (VC).
It is worth noting that the induction of vascular calcification by subcutaneous injection of VD3 for three days, followed by a six-day observation period, is a well-established and widely used model. This method reliably induces vascular calcification, as demonstrated by numerous studies. Our team has also successfully utilized this model in previous research and published several papers on the topic, including our previous works in Nature Communications (2016), Molecular Therapy - Nucleic Acids (2020), Experimental & Molecular Medicine (2021), and Experimental & Molecular Medicine (2024).
The relevant section in the Methods has been revised as follows to reflect these details.
To induce VC in mice models, 8- to 9-week-old male C57BL/6 mice were administered with VD3 solution subcutaneously at a dose of 150 μL/25 g (5 × 105 IU/kg/day) for 3 consecutive days, followed by a 6-day maintenance period to promote VC [20, 35]. VD3 solution was prepared by dissolving 14.56 mg cholecalciferol (C9759, Sigmal-Aldrich) in 70ul of 100% ethanol followed by mixing with cremophor (Alkamuls EL-620, Sigma-Aldrich) for 15 minutes at room temperature. The mixture was then combined with 6.2 ml of distilled water containing 250mg of dextrose for an additional 15 minutes at room temperature.
Below are the references
Price, P. A.; June, H. H.; Buckley, J. R.; Williamson, M. K., Osteoprotegerin inhibits artery calcification induced by warfarin and by vitamin D. Arterioscler Thromb Vasc Biol 2001, 21, (10), 1610-6.
Kwon, D. H.; Eom, G. H.; Ko, J. H.; Shin, S.; Joung, H.; Choe, N.; Nam, Y. S.; Min, H. K.; Kook, T.; Yoon, S.; Kang, W.; Kim, Y. S.; Kim, H. S.; Choi, H.; Koh, J. T.; Kim, N.; Ahn, Y.; Cho, H. J.; Lee, I. K.; Park, D. H.; Suk, K.; Seo, S. B.; Wissing, E. R.; Mendrysa, S. M.; Nam, K. I.; Kook, H., MDM2 E3 ligase-mediated ubiquitination and degradation of HDAC1 in vascular calcification. Nat Commun 2016, 7, 10492.
- Osteogenic differentiation description
Figure 3: The days of osteogenic differentiation should be described.
It has been described in the figure legends.
The day after the transfection, A10 cells were treated with 4 mM Pi for 4 days.
- Other matrix genes
Figures 3 and 4: Other matrix genes (collagen, osteocalcin, etc.) need to be quantified and analyzed to support the hypothesis.
During this revision, we performed qRT-PCR and added the result in Fig. 3d and Fig.4d
- Mechanism of effect of C/EBPβ on RUNX2
Figure 5: Explain about P1 and P2 promoters. Figure 5C does not illustrate the difference between P1 and P2 promoters, how C/EBPβ works, why C/EBPβ induces RUNX2 expression, and how Pi induces C/EBPβ.
As for the P1 and P2 promoters, we added following sentences to L132-137 and L183-187.
Runx2 expression is under control of two different promoters: the distal P1 and the proximal P2 promoter. The type II Runx2 driven by P1 promoter is predominantly expressed in bone progenitor cells and is upregulated during bone formation. In contrast, the type I Runx2 driven by P2 promoter is expressed in both osteogenic and non-osteogenic cells [23]. We explored whether the C/EBP regulates VC by activating the Runx2 promoter.
Runx2 has two distinct promoters: the distal P1 and the proximal P2 promoter. The Runx2 P1 promoter drives the type II transcript, encompassing the exon 1-8, and primarily regu-lates bone formation [29], in contrast to Runx2 P2 promoter drives the type I transcript, encompassing exon 2-8, and governs both osteogenic and non-osteogenic processes [30].
The detailed mechanism of how Pi induces C/EBPβ and how C/EBPβ activates the RUNX2 promoter will require further investigation in future studies. The primary significance of this study lies in identifying the potential role of C/EBPβ in promoting vascular calcification, which we believe is a meaningful contribution to the field.
To provide a more accurate explanation of our experimental results, we have revised the diagram in Figure 5C. The updated figure more clearly suggests the potential of C/EBPβ to activate the P2 promoter of RUNX2.
- Primer sequences
The sequences of qPCR primers should be provided.
We added following paragraph.
Pre-designed qPCR primer for Runx2 (Rn01512298_m1), ALP (Rn00677879_m1), and 18sRNA (Rn03928990_g1) were purchased from ThermoScientific Fisher (Germany). The sequences of primers used for PCR are as follows; C/EBPb, forward, 5’-CCAAGAAGACGGTGGACAA-3’, and reverse, 5’-TTGCGCATCTTGGCCTT-3’; OPN, forward, 5’- CCGTGAGGCCGCAGTTCTCC-3’, and reverse, 5’- CAGAGGGCACGCTCAGACGC-3’; Col1a1, forward, 5’-AGCATGTCTGGTTTGGAGAG-3’, and reverse, 5’- GTGATAGGTGATGTTCTGGGAG-3’; GAPDH, forward, 5’-TGCACCACCAACTGCTTA G-3’, and reverse, 5’-GATGCAGGG ATGATGTTC-3 ,
Reviewer 2 Report
Comments and Suggestions for Authors
Considering the clinical significance of vascular calcification during the progression of several chronic diseases and also in the context of aging, it is important to determine more reliable and targeted treatment strategies. The present work describes in detail the impact of C/EBPβ in the onset of vascular calcification, a protein that according to different recent studies may be involved in biological processes, including in bone and cartilage development and calcification. Even if the results were well described and the conclusions were in accordance with the findings, for a better understanding of C/EBPβ role in vascular calcification onset, it would be important to highlight the number of mice that were assessed and the renal function status of the monitored animals, considering the involvement of renal dysfunction in the development of vascular calcification.
Author Response
Renal failure in VD3-administered mice.
Considering the clinical significance of vascular calcification during the progression of several chronic diseases and also in the context of aging, it is important to determine more reliable and targeted treatment strategies. The present work describes in detail the impact of C/EBPβ in the onset of vascular calcification, a protein that according to different recent studies may be involved in biological processes, including in bone and cartilage development and calcification. Even if the results were well described and the conclusions were in accordance with the findings, for a better understanding of C/EBPβ role in vascular calcification onset, it would be important to highlight the number of mice that were assessed and the renal function status of the monitored animals, considering the involvement of renal dysfunction in the development of vascular calcification.
We sincerely thank the reviewer for their positive feedback and for providing insightful and constructive suggestions to improve our manuscript. We are particularly grateful for highlighting the importance of the study's clinical relevance and the detailed role of C/EBPβ in vascular calcification.
Regarding the number of mice used in the experiments, this information has been included in the Figure 2 legend for clarity.
It is indeed well-documented that excessive Vitamin D3 administration can lead to calcium overload, resulting in renal dysfunction. In our study, we observed phenotypic changes such as weight loss, reduced food intake, and decreased urine output following Vitamin D3 injection, which may indicate impaired renal function. Considering the established association between renal dysfunction and vascular calcification, it is plausible that the vascular calcification induced by Vitamin D3 in our study could be partially attributed to renal impairment.
However, we believe that the mechanism of Vitamin D3-induced vascular calcification itself falls outside the primary focus of this study, which aims to investigate the role of C/EBPβ in vascular calcification. We hope for the reviewer’s kind understanding regarding this distinction.
Thank you once again for your valuable comments, which have significantly contributed to the quality and clarity of our manuscript.
Round 2
Reviewer 1 Report
Comments and Suggestions for Authors
The authors revised the manuscript according to the reviewer's comments.